# Machine Learning Kreuzer–Skarke Calabi–Yau Threefolds

**Per Berglund**[a], **Ben Campbell**[a], **Vishnu Jejjala**[b]

[a]*Department of Physics and Astronomy, University of New Hampshire, Durham, NH 03824, USA*

[b]*Mandelstam Institute for Theoretical Physics, School of Physics, NITheCS, and CoE-MaSS,*
*University of the Witwatersrand, Johannesburg, WITS 2050, South Africa*
*E-mail:* Per.Berglund@unh.edu, Ben.Campbell@unh.edu,
vishnu@neo.phys.wits.ac.za

ABSTRACT: Using a fully connected feedforward neural network we study topological invariants of a class of Calabi–Yau manifolds constructed as hypersurfaces in toric varieties associated with reflexive polytopes from the Kreuzer–Skarke database. In particular, we find the existence of a simple expression for the Euler number that can be learned in terms of limited data extracted from the polytope and its dual.

## 1   Introduction

Superstring theory proposes that we inhabit a ten-dimensional spacetime. The Universe we observe, however, is plainly four-dimensional. One way to reconcile the mathematical consistency of quantum gravity with empirical facts about the world is for the predicted extra dimensions to be compact with a size determined by the string length. The energy scale needed to resolve such distances precludes experimental measurements of the extra dimensions. Compactification of the $E_8 \times E_8$ heterotic string on a Calabi–Yau threefold then presents a straightforward route to particle physics in the real world, or at the very least to a four-dimensional quantum field theory which has a non-Abelian gauge group and low energy chiral matter [1, 2]. In the simplest models, the number of generations of particles in the spectrum is $\frac{1}{2}|\chi(M)|$, with $\chi(M)$ being the Euler number of the compact Calabi–Yau manifold, $M$.

A Calabi–Yau space is a complex, Kähler manifold that admits a Ricci flat metric. (See [3, 4] for pedagogical reviews.) We must know the flat metric in order to compute features of the particle physics spectrum such as the Yukawa couplings and patterns of supersymmetry breaking. However, topology by itself carries us far, and following [5, 6], there are now thousands of semi-realistic constructions of the supersymmetric Standard Model from string compactification. The taxonomy of these spaces is involved. Some Calabi–Yau geometries arise as complete intersection hypersurfaces determined as the zero locus of polynomials in products of complex projective spaces, while others are hypersurfaces in toric varieties or are

elliptically fibered. These different species of Calabi–Yau manifolds have a non-trivial Venn diagram. In what follows we will focus on the hypersurface case.

More specifically, we study a class of $n$ complex dimensional spaces in which the Calabi–Yau manifold is described as an anticanonical divisor within an $(n + 1)$ complex dimensional toric variety arising from a reflexive polytope, $\Delta$, following the work of Batyrev [7]. Kreuzer and Skarke classified the lower dimensional cases, finding 4319 three-dimensional reflexive polyhedra [8] that give K3 (the Calabi–Yau twofold) and $473, 800, 776$ four-dimensional reflexive polyhedra [9, 10] leading to an unknown number of Calabi–Yau threefolds, while to date, there is only a partial list of the five-dimensional reflexive polyhedra [11] that give elliptically fibered Calabi–Yau fourfolds suitable for F-theory model building.

We will investigate the $n = 3$ dimensional Calabi–Yau spaces and the associated topological data, constructed from the class of reflexive four-dimensional $\Delta$. Starting from a given $\Delta$, we may algorithmically obtain a Calabi–Yau manifold from a consistent triangulation of the dual polytope, $\Delta^*$; see [12–14] for recent work detailing the explicit constructions of Calabi–Yau threefolds from polytopes. In particular, the Hodge numbers $h^{1,1}$ and $h^{2,1}$, which count, respectively, Kähler and complex structure deformations, as well as the Euler number, $\chi = 2(h^{1,1} - h^{2,1})$, are all explicitly given in terms of the combinatorial properties of $\Delta$ and its dual, $\Delta^*$ [7].[1]

Because the Kreuzer–Skarke catalogue of four-dimensional reflexive polytopes has nearly half a billion entries, it is securely within the realm of Big Data. Machine learning has emerged as a tool for analyzing such large data sets in string theory, in particular focusing on the topology of Calabi–Yau manifolds [16–19]. (See [20] for a review.) Indeed, some of the initial investigations in this direction looked at the complete intersection Calabi–Yau threefolds. There are 7890 such geometries [21], each of which is characterized by a configuration matrix that encodes the polynomial equations describing the complete intersection algebraic variety, $M$. Knowledge of this matrix is sufficient to predict the topological invariants $h^{1,1}$ and $\chi(M)$ [22–25]. Similar investigations have been carried out for the complete intersection Calabi–Yau fourfolds [26–28]. Since the calculations scale polynomially with the size of the configuration matrix instead of doubly exponentially as would be expected from sequence chasing or Gröbner basis methods in computational algebraic geometry, one of the conclusions of [22, 23] is that there may be more efficient ways to calculate the Hodge numbers. In this paper, we machine learn topological invariants of the Calabi–Yau threefolds constructed as

---

[1] The Hodge data do not by any means uniquely define the Calabi–Yau manifold. For example, there are nearly one million polyhedra with $(h^{1,1}, h^{2,1}) = (27, 27)$. The statistics of these polytopes is discussed in [15].

hypersurfaces in toric varieties from a limited amount of data describing four-dimensional polytopes and dual polytopes taken from the Kreuzer–Skarke list [10, 29].[2]

Finally, it is well known that artificial intelligence methods excel at finding associations between features in data. In certain cases, neural network based curve fitting has facilitated the search for new analytic formulæ. Instances of this occur in examining line bundle cohomology on surfaces and on Calabi–Yau threefolds [31–34] and in analyzing the topological invariants of knots [35–40]. Here we use the machine learning results concerning reflexive polytopes to deduce new analytic expressions for topological invariants.

The organization of this paper is as follows. In Section 2, we review the Kreuzer–Skarke data set. In Section 3, we detail the machine learning methods we employ. In Section 4, we describe the machine learning predictions of toric Calabi–Yau threefold Hodge numbers from four-dimensional reflexive polytope data. In Section 5, we present a new analytic expression for the Euler number. In Section 6, we provide a prospectus for future work in this direction.

## 2  Kreuzer–Skarke Data

Kreuzer and Skarke tabulated all $473,800,776$ reflexive polyhedra in four dimensions [9, 10]. Starting from any one of these polytopes, $\Delta$, we can construct a four-dimensional toric variety $X_\Delta$ in which the anticanonical hypersurface is a (possibly) singular Calabi–Yau variety realized as a generic section of the anticanonical bundle of $X_\Delta$ [7]. In general, each such hypersurface may admit several maximal projective crepant partial desingularizations each associated to a given triangulation of the dual polytope, $\Delta^*$, with distinct Stanley–Reisner rings corresponding to different geometries. Furthermore, the same Calabi–Yau threefold could as well arise from triangulating different four-dimensional reflexive polytopes. In one lower dimension, this is what happens for K3: whereas there are $4319$ reflexive polytopes in three dimensions, any two complex analytic K3 surfaces are diffeomorphic as smooth four manifolds [41]. The Calabi–Yau twofold is essentially unique. In contrast, we do not have even a rough enumeration of the corresponding Calabi–Yau threefolds. It is, however, likely that this number is well in excess of the half a billion reflexive polytopes.

We focus our attention in this paper on a subset of the topological properties of the Calabi–Yau spaces obtained from the polytope data. For the class of four-dimensional reflexive polytopes there are $30,108$ unique pairs of Hodge numbers $(h^{1,1}, h^{2,1})$ of the associated Calabi–Yau manifold. This sets a lower bound on the number of Calabi–Yau threefolds. Our goal is to compute these topological invariants of a Calabi–Yau threefold using only minimal information about the polytopes.

---

[2]See also [30].

Let us briefly review what the polytope data provide. For toric geometry, we must specify a reflexive polytope $\Delta$ and its dual $\Delta^*$. A polytope in $\mathbb{R}^4$ is the convex hull of finitely many lattice points, the number of which is denoted by $l(\Delta)$. That is to say, we specify the vertices with integer valued vectors. Pairs of neighboring vertices define an edge, collections of edges define a face, etc. Adopting the nomenclature of [8], a polytope is said to have the *interior point (IP)* property when it contains the origin as its sole interior point with integer coordinates. The *dual (polar) polytope* is defined as

$$\Delta^* = \{v \in \mathbb{R}^4 \mid \langle m, v \rangle \geq -1 \; \forall\, m \in \Delta\} \,, \tag{2.1}$$

where the inner product $\langle m, v \rangle$ is calculated in $\mathbb{R}^4$. The polytope is *reflexive* when the vertices of $\Delta^*$ are specified by integer vectors, and $\Delta^*$ shares the IP property, from which it follows that $(\Delta^*)^* = \Delta$. Furthermore, the convex hull of $\Delta^*$ consists of $l(\Delta^*)$ points. The Calabi–Yau hypersurface $M$ is constructed as a generic section of the anticanonical bundle, $-K_X$, on $X_\Delta$, and explicitly given by the following vanishing condition

$$0 = \sum_{m \in \Delta} a_m \prod_i x_i^{\langle m, v_i^* \rangle + 1} \,, \tag{2.2}$$

where the $v_i^*$ are vertices of $\Delta^*$, $x_i$ are coordinates on the toric variety, $X_\Delta$, and $a_m$ are coefficients that parameterize the complex structure of $M$. Exchanging $\Delta$ and $\Delta^*$ provides an analogous construction of the mirror Calabi–Yau $W$, for which $h^{1,1}$ and $h^{2,1}$ are interchanged:

$$0 = \sum_{m \in \Delta^*} b_m \prod_i y_i^{\langle m, v_i \rangle + 1} \,, \tag{2.3}$$

where now the $v_i$ are vertices of $\Delta$, $y_i$ are coordinates on the "mirror" toric variety, $X_{\Delta^*}$, and $b_m$ are coefficients that parameterize the complex structure of the mirror manifold $W$.

In what follows, we as well introduce the scaled up polytopea $k\Delta$ and $k\Delta^*$, where $k = 2$ is the integer scale factor. Note that neither $k\Delta$ nor $k\Delta^*$ are reflexive polyhedra in their own right. To define the scaled up objects, we simply multiply the vectors that define the vertices of $\Delta$ and $\Delta^*$ by $k = 2$, with $l(2\Delta)$ and $l(2\Delta^*)$ the number of points in the convex hull of the scaled up polytopes, respectively. Remarkably, we find that specifying $(l(\Delta), l(\Delta^*), l(2\Delta), l(2\Delta^*))$, *i.e.*, the number of lattice points on $\Delta$, and the corresponding data for $\Delta^*$, $2\Delta$, and $2\Delta^*$, is sufficient to machine learn topological invariants of the Calabi–Yau threefold.

Our analysis contrasts with machine learning efforts on complete intersection Calabi–Yaus. There, the configuration matrix provides the complete information about how the geometry is realized. Here, we are not even providing the full polytope data, *viz.*, the vectors that identify the vertices of $\Delta$ (and by duality, the vertices of $\Delta^*$). To specify the geometry, we must go even further by triangulating the polytope.

## 3 Machine Learning Methods

A fully connected feedforward neural network accomplishes a non-linear regression. To an input vector $\boldsymbol{v}$, we associate the output

$$f_\theta(\boldsymbol{v}) = \left|\left| L^{(n)}(\sigma_{n-1}(L^{(n-1)}(\ldots \sigma_1(L^{(1)}(\boldsymbol{v}))))) \right|\right|_1 , \tag{3.1}$$

where

$$\boldsymbol{v}^{(k)} = \sigma_k(L^{(k)}(\boldsymbol{v}^{(k-1)})) , \qquad L^{(k)}(\boldsymbol{v}^{(k-1)}) = W^{(k)} \cdot \boldsymbol{v}^{(k-1)} + \boldsymbol{b}^{(k)} , \tag{3.2}$$

with $\boldsymbol{v}^{(0)} = \boldsymbol{v}$. This is an $n$ layer neural network. The $\theta$ collectively denotes a set of hyperparameters, which are the matrix elements of the weight matrices $W^{(k)}$ and the components of the bias vectors $\boldsymbol{b}^{(k)}$. If $W^{(k)}$ is an $m_k \times m_{k-1}$ matrix, there are $m_k$ neurons in the $k$-th hidden layer. The functions $\sigma_k$ implement the non-linearity elementwise on $L^{(k)}(\boldsymbol{v}^{(k-1)})$. Suppose we know that we should ascribe the result $\phi_i$ to an input vector $\boldsymbol{v}_i$. The hyperparameters are randomly initialized and set in training by optimizing a loss function

$$\Upsilon(\theta) = \sum_i \rho(\boldsymbol{v}_i) d(f_\theta(\boldsymbol{v}_i), \phi_i) \tag{3.3}$$

over a collection of input vectors, where the summands in (3.3) are a suitable measure of the difference between the neural network prediction $f_\theta(\boldsymbol{v}_i)$ and the true answer $\phi_i$, weighted by some density.

The models in these experiments are implemented using `Julia 1.6.2` [42] and the `Flux` package [43, 44]. Each model consists of a neural network with five hidden layers with 300 neurons per layer. We use rectified linear unit (ReLU) activation functions so that $\sigma(x) = \max(0, x)$. Optimization is enacted via the Adam algorithm [45] and a logit cross entropy loss function. The initial learning rate is $\eta = 10^{-3}$, and a learning rate schedule is set such that after eight epochs with no improvement the learning rate is halved until it is less than $\eta = 10^{-5}$, after which $\eta$ is fixed. (The learning rate is a parameter of the neural network that determines how much to change the model in response to the estimated error at each iteration.) The maximum number of training epochs, or passes through the entire training data, is set at 50 as longer training shows no improvement.

Of the roughly half-billion four-dimensional polytopes from the Kreuzer–Skarke database, a random sample of one million polytopes was selected, as shown in Figure 1. The model was trained on 80% of the polytopes and tested on the complementary 20% of the data set. This procedure was then repeated 100 times, where the total sample is shuffled and split before each model is trained; thus each model was trained and tested on a different selected subset of the originally sampled one million polytopes. The input data is given by the input vector

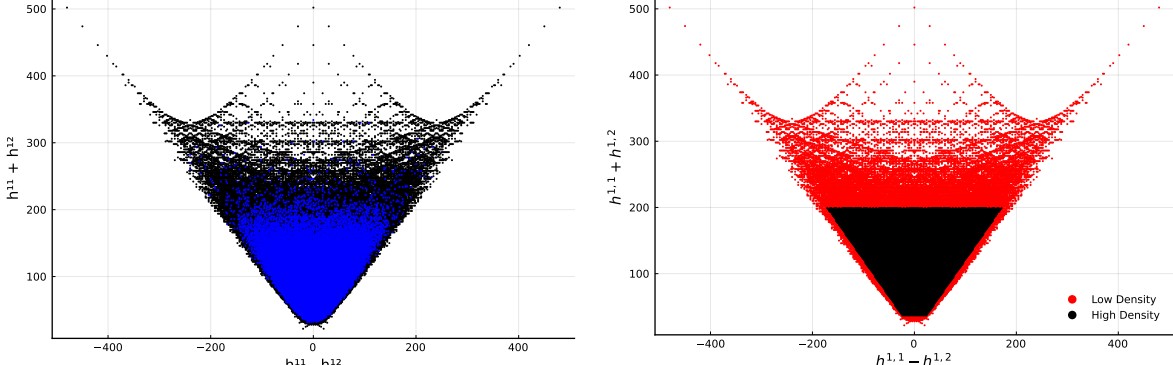

**Figure 1**. Distribution of randomly sampled data as a plot of $h^{1,1} - h^{2,1}$ vs. $h^{1,1} + h^{2,1}$. Blue represents an $(h^{1,1}, h^{2,1})$ pair for which a polytope has been randomly selected.

**Figure 2**. Distribution of boundary data as a plot of $h^{1,1} - h^{2,1}$ vs. $h^{1,1} + h^{2,1}$. Red represents an $(h^{1,1}, h^{2,1})$ pair for which all polytopes with this data are sampled.

$\boldsymbol{v} = (l(\Delta), l(\Delta^*), l(2\Delta), l(2\Delta^*))$, that is the number of lattice points on $\Delta$, its dual $\Delta^*$, as well as the twice scaled up polytopes, $2\Delta$, and $2\Delta^*$, respectively, data which are calculated from any given Kreuzer–Skarke polytope $\Delta$. The set of hyperparameters is chosen to maximize accuracy in training $\chi = 2(h^{1,1} - h^{2,1})$. No appreciable difference in accuracy was found when different sets of optimal hyperparameters were obtained from training to predict $h^{1,1}$, $h^{2,1}$, and $h^{1,1} + h^{2,1}$.

The mirror symmetric plot of $h^{1,1} + h^{2,1}$ vs. $h^{1,1} - h^{2,1}$ is densely populated in the center. As a result, the majority of the randomly selected data comes from this region. The machine learning trials were therefore repeated using only data on the boundary, where the boundary is approximated by four linear conditions for ease, see Figure 2. Models were trained to predict $\chi$, $h^{1,1}$, $h^{2,1}$, and $h^{1,1} + h^{2,1}$ with the same 80% training to 20% testing ratio as in the original set of experiments.[3] The data are again split and shuffled before each model was trained. The set of hyperparameters was varied, though no considerable change in performance was obtained compared to the values used for the original randomly selected data set. However, the maximum number of training epochs was increased to 100 as this allowed for improvement in accuracy as opposed to the randomly sampled data.

## 4   Numerical Results

A plot of $h^{1,1} - h^{2,1}$ vs. $h^{1,1} + h^{2,1}$ for the randomly selected data and the chosen boundary data are shown in Figures 1 and 2, respectively. Models are trained to predict $\chi$, $h^{1,1}$, $h^{2,1}$,

---

[3]The case $h^{1,1} + h^{2,1}$ was included as the natural "dual" expression to $\chi$, though in fact the testing accuracy was the lowest of the different labels used.

and $h^{1,1} + h^{2,1}$ for each data set. In a given trial, the mean absolute error for $h^{1,1}$ is defined by taking the expectation value

$$\delta_{h^{1,1}} = \langle |h^{1,1}_{\text{predicted}} - h^{1,1}_{\text{true}}| \rangle \tag{4.1}$$

over the test data set, which is complementary to the collection of vectors used in training. Similar expressions compute $\delta_{h^{2,1}}$, $\delta_{h^{1,1}+h^{2,1}}$, and $\delta_\chi$. The mean relative absolute error $\hat{\delta}$ normalizes by dividing the difference between the predicted and true values by the true value. When we compute the mean absolute relative error for $\chi$, we normalize by $|\chi_{\text{true}}|$ and exclude the cases where this vanishes.

For predicting $\chi$, the models averaged an accuracy of $97.36\% \pm 1.42\%$ and an absolute error of $0.1746 \pm 0.1941$ when trained and tested on the randomly sampled data and an accuracy of $96.02\% \pm 1.24\%$ and absolute error of $0.2560 \pm 0.0702$ on the boundary data. Models trained and tested on the boundary data did better overall for the remaining trials. For $h^{1,1}$, models trained on the randomly sampled data averaged an accuracy of $46.89\% \pm 0.91\%$ and an absolute error of $0.7099 \pm 0.0966$ while models trained on the boundary data averaged an accuracy of $75.35\% \pm 1.08\%$ and absolute error of $0.3142 \pm 0.0494$. Similar results hold for $h^{2,1}$ with models averaging an accuracy of $46.74\% \pm 1.03\%$ and an absolute error of $0.7262 \pm 0.0896$ when trained on the randomly sampled data while averaging and accuracy of $75.48\% \pm 0.78\%$ and absolute error of $0.3131 \pm 0.0632$. Finally, for predicting $h^{1,1} + h^{2,1}$, models averaged an accuracy of $32.64\% \pm 0.27\%$ and absolute error of $1.464 \pm 0.046$ for the randomly sampled data and an accuracy of $67.77\% \pm 1.60\%$ and absolute error of $0.7122 \pm 0.0649$. The accuracies are sometimes low, but this comparison queries whether the prediction exactly matches the true value. The absolute errors indicate that the wrong predictions of the neural network are not very wrong. The results for models trained on randomly sampled and boundary data are enumerated in Tables 1 and 2, respectively. The standard deviations are obtained from computing the variance over 100 runs.

Confusion matrices for models trained and tested on the randomly sampled data and trained and tested on the boundary are shown in Figures 3 and 4, respectively. The model used to generate the confusion matrix for the randomly sampled data achieved an accuracy of $99.25\%$ and mean absolute error of $0.0123$ on the randomly sampled testing data while the model used to generate the confusion matrix for the boundary data had an accuracy of $96.32\%$ and mean absolute error of $0.1339$. Models trained on the randomly selected data are also evaluated on the boundary data to see how well they extrapolate outside of the central region of Figure 1. Predicting $\chi$ from boundary data with a model trained on randomly sampled data resulted in an accuracy of $78.43\%$ and average absolute error of $1.19$.

| Label | Accuracy (%) | Absolute Error | Relative Absolute Error |
|:---:|:---:|:---:|:---:|
| $\chi$ | $97.36 \pm 1.42$ | $0.1746 \pm 0.1941$ | $0.0049 \pm 0.0099$ |
| $h^{1,1}$ | $46.89 \pm 0.91$ | $0.7099 \pm 0.0966$ | $0.0222 \pm 0.0029$ |
| $h^{2,1}$ | $46.74 \pm 1.03$ | $0.7262 \pm 0.0896$ | $0.0227 \pm 0.0026$ |
| $h^{1,1} + h^{2,1}$ | $32.64 \pm 0.27$ | $1.464 \pm 0.046$ | $0.0214 \pm 0.0006$ |

**Table 1**. Mean accuracy, absolute error, and relative absolute error for model predictions on each label trained and tested on the randomly sampled data. Averages and standard deviations are taken over 100 models for each label. The total data is shuffled before being split into 80% training and 20% testing sets for each model.

| Label | Accuracy (%) | Absolute Error | Relative Absolute Error |
|:---:|:---:|:---:|:---:|
| $\chi$ | $96.02 \pm 1.24$ | $0.2560 \pm 0.0702$ | $0.0035 \pm 0.0018$ |
| $h^{1,1}$ | $75.35 \pm 1.08$ | $0.3142 \pm 0.0494$ | $0.0090 \pm 0.0008$ |
| $h^{2,1}$ | $75.48 \pm 0.78$ | $0.3131 \pm 0.0632$ | $0.0089 \pm 0.0009$ |
| $h^{1,1} + h^{2,1}$ | $67.77 \pm 1.60$ | $0.7122 \pm 0.0649$ | $0.0075 \pm 0.0007$ |

**Table 2**. Mean accuracy, absolute error, and relative absolute error for model predictions on each label trained and tested on the boundary data. Averages and standard deviations are taken over 100 models for each label. The total data is shuffled before being split into 80% training and 20% testing sets for each model.

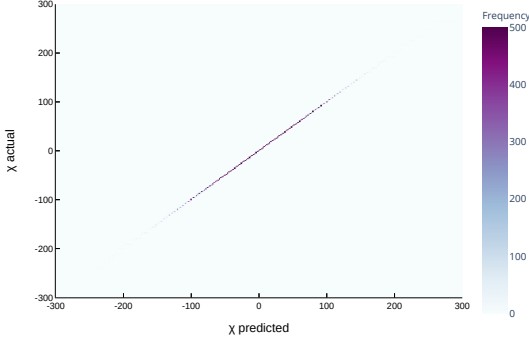

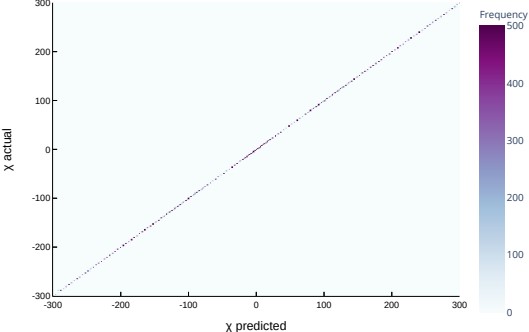

**Figure 3**. Confusion matrix for model trained on randomly sampled data evaluated on randomly sampled data. This model achieved an accuracy of 99.25% and mean absolute error of 0.0123. The range has been cropped to $\chi \in [-300, 300]$ as the majority of the data is in this interval.

**Figure 4**. Confusion matrix for model trained on boundary data evaluated on boundary data. This model achieved an accuracy of 96.43% and a mean absolute error of 0.1339. The range has been cropped to $\chi \in [-300, 300]$ for comparison with Figure 3.

The ability of the neural network to predict the topological invariants with some accuracy suggests the existence of approximate analytic formulæ. To test this hypothesis, we considered a random sample of $200,000$ four-dimensional reflexive polytopes and their mirrors from the Kreuzer–Skarke list. Taking the number of points in $l(\Delta)$, the number of points in $l(\Delta^*)$, the number of points in $l(2\Delta)$, and the number of points in $l(2\Delta^*)$, we performed a linear regression on these data to fit expressions for $h^{1,1}$ and $h^{2,1}$ using `Mathematica 12.3.1` [46]. Selecting half of these polytopes and their mirrors at random and repeating the regression 100 times, we find:

$$h^{1,1} = -(3.33 \pm 0.05) - (3.471 \pm 0.005)l(\Delta) + (5.529 \pm 0.005)l(\Delta^*) +$$
$$+(0.4176 \pm 0.0007)l(2\Delta) - (0.5824 \pm 0.0007)l(2\Delta^*) , \quad (4.2)$$
$$h^{2,1} = -(3.33 \pm 0.05) + (5.529 \pm 0.005)l(\Delta) - (3.471 \pm 0.005)l(\Delta^*) +$$
$$-(0.5824 \pm 0.0007)l(2\Delta) + (0.4176 \pm 0.0007)l(2\Delta^*) . \quad (4.3)$$

Plugging in the values of $l(\Delta)$, $l(\Delta^*)$, $l(2\Delta)$, and $l(2\Delta^*)$ into (4.2) and rounding to the nearest integer predicts $h^{1,1}$ correctly 44% of the time. The prediction is off by one another 44% of the time. In total, the mean of the absolute value of the error in the prediction is $0.70 \pm 0.85$ on the full data set. Similarly, the prediction of $h^{2,1}$ in (4.3) is correct 46% of the time and off by one 42% of the time. In total, the mean of the absolute value of the error in the prediction is $0.70 \pm 1.09$.

If we fit instead to $h^{1,1} + h^{2,1}$ and $h^{1,1} - h^{2,1}$, we find

$$h^{1,1} + h^{2,1} = -(6.64 \pm 0.01) + (2.06 \pm 0.01)\Big(l(\Delta) + l(\Delta^*)\Big)$$
$$-(0.165 \pm 0.001)\Big(l(2\Delta) + l(2\Delta^*)\Big) , \quad (4.4)$$
$$\frac{1}{2}\chi = h^{1,1} - h^{2,1} = 9\Big(l(\Delta^*) - l(\Delta)\Big) + \Big(l(2\Delta) - l(\Delta^*)\Big) . \quad (4.5)$$

The formula for the sum of the Hodge numbers (4.4) is exactly correct 23% of the time and off by one 38% of the time. The mean of the absolute value of the error in the prediction is $1.45 \pm 1.39$. The formula for the difference of the Hodge numbers (4.5), on the other hand, is exact. The coefficients in the fit are integer and have zero error. The prediction is perfect for every polytope. We will now derive this expression as a new analytic formula for the Euler number.

## 5   Analytic Formulæ

In what follows, we will focus on the so called stringy topological data as originally introduced by Batyrev in the context of reflexive polytopes, $\Delta$, and the associated toric varieties, $X_\Delta$ [7,

[47, 48].[4] The main idea is that certain topological invariants, including the Hodge numbers and in particular the Euler number, are independent of the choice of the so called crepant desingularization of $X_\Delta$ [50, 51].

Starting from the Ehrhart series,

$$P_\Delta(t) = \sum_{k \geq 0} l(k\Delta)t^k , \qquad (5.1)$$

we use that we can rewrite this as

$$P_\Delta(t) = \frac{\Phi(t)}{(1-t)^{n+1}} , \qquad (5.2)$$

where $n = \dim \Delta$. Here, we have introduced the Ehrhart polynomial,

$$\Phi(t) = \sum_{i=0}^{n} \psi_i(\Delta)t^i , \qquad (5.3)$$

where $\psi_i(\Delta)$ encode topological information about the toric variety $X_\Delta$. For a reflexive polytope, the $\psi_i(\Delta)$ satisfy $\psi_{n-i}(\Delta) = \psi_i(\Delta)$, where we in addition know that $\psi_0(\Delta) = 1$ since $l(0\Delta) = 1$. By expanding the geometric series we can determine the $\psi_i(\Delta)$ in terms of the $l(k\Delta)$. In particular, from the above it then follows that $\psi_1(\Delta) = l(\Delta) - n - 1$ and $\psi_2(\Delta) = l(2\Delta) - (n+1)l(\Delta) + n(n+1)/2$.

Libgober and Wood showed that there exists an identity relating $c_1(X)c_{n-1}(X)$ and the Hodge numbers, $h^{p,q}(X)$ for $X$ a smooth projective variety [52]. This was generalized by Batyrev, *et al.* to any toric variety $X_\Delta$ associated to a reflexive polytope $\Delta$ in terms of the combinatorial data of $\Delta$ [7, 47, 48],

$$\sum_{i=0}^{n} \psi_i(i - \frac{n}{2})^2 = \frac{n}{12}d(\Delta) + \frac{1}{6} \sum_{\substack{\theta \in \Delta \\ \dim \theta = n-2}} d(\theta)d(\theta^*) . \qquad (5.4)$$

Here $d(\theta)$ refers to the so called degree of the face $\theta \in \Delta$ and is given by $d(\theta) = (n-2)!\text{Vol}(\theta)$. In particular, $d(\Delta)$ is the volume of $\Delta$, up to the factor of $n!$. For the case $n = 3$, this leads to the statement that all K3 surfaces have $\chi = 24$:

$$\chi(M) = \sum_{\substack{\theta \in \Delta \\ \dim \theta = 1}} d(\theta)d(\theta^*) = 24 . \qquad (5.5)$$

In writing this expression, we have used the fact that the Euler number $\chi$ for the case of a Calabi–Yau hypersurface $M$ defined by the anticanonical divisor $-K_X$ associated to the

---

[4]Related work on calculating the stringy Chern classes, $c_i(X)$, in terms of the combinatorial data for more general polytopes $\Delta$ can be found in [49].

anticanonical bundle $\mathcal{O}(-K_X)$ of the toric variety $X_\Delta$ is given by [48]

$$\chi(M) = \int_X c_1(X)c_2(X) = \sum_{\substack{\theta \in \Delta \\ \dim \theta = 1}} d(\theta)d(\theta^*) \ , \tag{5.6}$$

generalizing the standard result for the smooth case, *cf.* [53].

In $n = 4$ dimensions, the generalized Libgober–Wood identity for $X_\Delta$ can be shown to be given by [48]

$$12\left(l(\Delta) - 1\right) = 2d(\Delta) + \sum_{\substack{\theta \in \Delta \\ \dim \theta = 2}} d(\theta)d(\theta^*) \ , \tag{5.7}$$

and similarly for $\Delta^*$,

$$12\left(l(\Delta^*) - 1\right) = 2d(\Delta^*) + \sum_{\substack{\theta^* \in \Delta^* \\ \dim \theta^* = 2}} d(\theta^*)d(\theta) \ . \tag{5.8}$$

Analogously to the $n = 3$ case, we can write

$$\chi(M) = \int_X \left[c_1(X)c_3(X) - c_1^2(X)c_2(X)\right] = \sum_{\substack{\theta \in \Delta \\ \dim \theta = 1}} d(\theta)d(\theta^*) - \sum_{\substack{\theta \in \Delta \\ \dim \theta = 2}} d(\theta)d(\theta^*) \ . \tag{5.9}$$

Using that $\dim \theta^* = n - 1 - \dim \theta$, it then follows that for a reflexive polytope in $n = 4$ dimensions, we can rewrite the right hand side of the expression for $\chi(M)$ in terms of the difference of the Libgober–Wood identities for $\Delta$ and $\Delta^*$, respectively,

$$\chi(M) = 12\left(l(\Delta^*) - l(\Delta)\right) + 2\left(d(\Delta) - d(\Delta^*)\right) \ . \tag{5.10}$$

Finally, we can express this in terms of the number of points, $l(2\Delta)$, $l(2\Delta^*)$, of the twice scaled up polytopes $2\Delta$ and $2\Delta^*$, respectively, using that [54]

$$d(\Delta) = 2 + l(2\Delta) - 3l(\Delta) \ , \qquad d(\Delta^*) = 2 + l(2\Delta^*) - 3l(\Delta^*) \ , \tag{5.11}$$

and hence,

$$\chi(M) = 2\left(l(2\Delta) - l(2\Delta^*)\right) + 18\left(l(\Delta^*) - l(\Delta)\right) \ . \tag{5.12}$$

This agrees with the expression (4.5) found by linear regression. As a simple example, let us consider $\Delta$ describing the Newton polyhedron associated with the degree five hypersurface $M$ in $X = \mathbb{P}^4$, for which we know $(h^{1,1}, h^{2,1}) = (1, 101)$. Using the Kreuzer–Skarke database one finds that $l(\Delta) = 126$, $l(\Delta^*) = 6$ [10], and furthermore calculates that $l(2\Delta) = 1001$ and $l(2\Delta^*) = 21$. The above expression for the Euler number correctly gives $\chi(M) = -200$.

## 6    Discussion and Prospectus

Neural networks are universal approximators [55, 56]. This means that the output of the first layer of a finite width neural network can approximate any suitably well behaved function on a compact subset of $\mathbb{R}^n$. We have obtained an analytic expression for the Euler number $\chi(M)$ from data about points on the reflexive polytope $\Delta$, its dual and the scaled up versions of these, respectively, where $M$ is given by the anticanonical divisor $-K_X$ of the toric variety $X_\Delta$ associated to $\Delta$. Since the data, $l(\Delta)$, $l(\Delta^*)$, $l(2\Delta)$, and $l(2\Delta^*)$, are integer valued, there are many suitable functions. With a simple architecture, the neural network approximates one of these. The better than 96% accuracy further suggests that these results should be analyzed from the perspective of probably approximately correct learning [57]. The lower accuracy of the predictions of $h^{1,1}(M)$, $h^{2,1}(M)$, and the sum $h^{1,1} + h^{2,1}(M)$ indicates that there probably is not a similarly simple formula for these topological invariants based on the minimal data we have provided.

The accuracy for predicting $h^{1,1}$ and $h^{2,1}$ is considerably higher for data along the boundary. This may be explained by the fact that there do exist exact expressions for the individual Hodge numbers for special classes of reflexive polytopes, $\Delta$, such that $h^{2,1} = l(\Delta) - 5$, and similarly for the dual polytopes, $\Delta^*$, with $h^{1,1} = l(\Delta^*) - 5$.[5] These classes of polytopes do indeed occur to a larger degree along the boundary.

It would be interesting to determine what other data we can add to get more precise predictions of the individual Hodge numbers. Clearly, knowing the lattice points of the polytope is enough to compute these quantities, but perhaps we can get away with supplying less information in the input. It would also be interesting to test whether quantities like the Kähler cone and the Mori cone can be machine learned.

Calabi–Yau manifolds are an important testbed for applying machine learning to problems in string theory. We have focused our attention in this paper on the Hodge numbers. The success of machine learning methods in this endeavor may point to structural features in the data set taken as a whole. Based on [58], Reid famously conjectured that the moduli space of Calabi–Yau threefolds is connected through conifold transitions and that any Calabi–Yau geometry may be obtained from the small resolution of degenerations within this family [59]. Perhaps this insight is at the heart of the machine learned relationships. (Similar speculations appear in [60].)

---

[5]The latter case corresponds to the statement that $h^{1,1}(M) = \dim \text{Pic}(X_\Delta)$, the Picard number of the ambient space.

Finally, we would like to go beyond topology to geometry. First steps in this direction have constructed numerical Ricci flat metrics on Calabi–Yau threefolds via machine learning [61–66]. In line with Reid's fantasy, we would as well like to understand geometric transitions from the perspective of machine learning. This is work in progress [67].

## Acknowledgments

We thank Damián Mayorga Peña, Challenger Mishra, and in particular Tristan Hübsch for discussions, and Philip Candelas for `Mathematica` code used to analyze reflexive polytopes. PB and BC are supported in part by the Department of Energy grant DE-SC0020220. VJ is supported by the South African Research Chairs Initiative of the Department of Science and Technology and the National Research Foundation and by the Simons Foundation Mathematics and Physical Sciences Targeted Grant, 509116.

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
