# Peer review of "Machine Learning Kreuzer--Skarke Calabi--Yau Threefolds"

_SciPost Physics_

## Round 1 · Referee Report · Anonymous (Referee 1) · 2022-5-9

Report

The authors apply machine learning techniques to the database of four-dimensional reflexive polytopes. Each of these (approximately 500 million) polytopes gives rise to a family of Calabi-Yau threefolds characterized by a specific pair of Hodge numbers (there are roughly 30,000 distinct pairs). The goal is to predict the Hodge numbers solely on the basis of the numbers of lattice points in the following four polytopes: the given one, the double of the given polytope, the dual polytope and the double of the dual polytope.

Two types of training sets are used: one of them corresponds to randomly picked entries in the database (this favors moderate Hodge number pairs), and the other one to the boundary of the Hodge number plot, which corresponds to rarely occurring Hodge number pairs. The choice of training data makes quite a difference to the performance of the neural network: in the first case, single Hodge numbers are correctly predicted in slightly less than half of the cases and the sum in roughly one third of all cases, but these numbers improve to approximately 3/4 and 2/3, respectively, for the training data in the boundary. The most striking result concerns the difference of the two Hodge numbers (half the Euler number). This number is correctly predicted in more than 95% of the cases for either training set. The authors draw the obvious (correct) conclusion that this hints at the fact that the Euler characteristic is actually a function of the input data. This is corroborated with the help of linear regression, which provides a simple linear function that determines the Euler characteristic in terms of the four given integers. The authors then proceed to prove the formula they found in this way. The proof is far from being elementary, involving Ehrhart series and Libgober-Wood identities.

The paper is well written. It certainly rises above the large number of machine-learning applications to any reasonably large database, by actually finding a new (to the best of my knowledge) result and a non-trivial proof for it. I am strongly in support of publication.

Requested changes

There is a typographical error in the central formula (4.5): The second occurrence of l(delta*) should be replaced by l(2 delta*).

---

## Round 1 · Referee Report · Harold Erbin (Referee 2) · 2022-5-15

Strengths

  1. New application of machine learning techniques to Calabi-Yau manifolds.
  2. Excellent introduction.

Weaknesses

  1. Weak motivation for using neural networks compared to linear regression.
  2. A better comparison with previous analytic formulas should be provided.
  3. The authors do not provide the code.

Report

This paper computes Hodge numbers and linear combinations thereof for Kreuzer-Skarke Calabi-Yau 3-folds using neural networks. This is an interesting prospect as it extends the range of applications of machine learning for computing properties of Calabi-Yau manifolds and string compactifications. In particular, the input data does not contain all the information defining the geometry. This submission does not meet the criteria of Physics, but does meet those of Physics Core, where it could be published. Before, several points need to be clarified, especially, the fact that neural networks do not seem to be needed to reach the conclusions of the paper.

(see attached document for PDF version of full report)

Requested changes

  1. p. 5, §2: What dictated the choice of architecture? There seems to be some margin of improvement for some results, so why not consider more complicated architectures (convolutions, dropout, early stopping…)?

  2. p. 5, §2: The structure of the predictions should be made clearer. It is said that the activation functions are ReLU, but the loss function is the logit cross-entropy. Does it mean that the last activation function is in fact a sigmoid/softmax? Is there a single network predicting all quantities at the same time (multi-tasking) -- which seems to be the case from sec. 1, §1 --, or four independent networks? In both cases, if classification is used, how is the outputs represented (categorical or one-hot)?

Note that using a classification task means that one assumes knowledge of the boundary, so this assumption must be clearly stated.

  1. p. 5, last §: The usual ML methodology requires setting aside a test set and performing training and hyperparameter tuning using only a training set (split as training and validation, or using cross-validation as in the paper), evaluating the performance on the test set at the end. Since the authors have used the same set for all steps, there is a risk of overfitting. Hence, the authors should perform again the analysis with a proper training/test split + $k$-fold cross-validation on the training set only (statistics on the predictions for the test set can still be obtained using the $k = 100$ cross-validated models).

Moreover, I don't understand why the authors are considering only a subset of $10^6$ polytopes instead of using all the remaining polytopes as a test set.

(This would still allow studying general random data and data only on the boundary: in the second case, first exclude the data in the center, then do the same thing as above.)

  1. p. 7, §2: I am surprised that the accuracy for all quantities but $\chi$ are so low ($32\%$ to $46\%$). The small error seems to indicate that many predictions are off by only a little, but could better architectures (see above) improve these results?.

  2. p. 7, last §: I don't understand why the accuracy are higher in this paragraph compared to the previous paragraph? The change seems higher than the previously quoted deviation of the accuracy. Moreover, you could get the confusion matrix directly from the network trained in the previous paragraph. Is it because you trained a new single-tasking network to predict only $\chi$?

  3. p. 9, §1: I don't think that $46\%$ accuracy is a sufficiently high for “[suggesting] the existence of approximate analytic formulae”, especially in the presence of possible overfitting (see previously).

I am also intrigued by the fact that the linear formulas in (4.2) and (4.3) are as precise as the results from the neural network (both accuracy and absolute errors are almost equal). Do they work both for the same polytopes or on different ones? In the first case, what is the use of neural networks if linear regression performs as good? It would have been more traditional from an ML perspective to first do a linear regression, and then try to beat it with neural networks.

Could you also comment by comparing with known analytic formulas, for example corollary 4.5.1 in [7]?

  1. p. 9, §2: The same comment holds for (4.5): why bother using neural networks if linear regression works better? There can be a case for $h^{1,1} + h^{2,1}$ since linear regression is significantly less performant in that case, but then using a neural network for only that quantity could give better performance than the ones obtained previously.

  2. p. 11: Formula (5.9) was given in Theorem 4.5.3 from [7], and it seems that (5.12) follows quite directly from the identities (5.7) and (5.8) from [48], and (5.11) from [54]. This looks like a rewriting of known formulas in a slightly different form, and it does not seem that ML did really help. Finally, from the previous point, it is not clear what's the role of using neural networks if linear regression is sufficient.

Attachment

---

## Editorial Decision

awaiting_resubmission